# Is Glial Dysfunction the Key Pathogenesis of *LRRK2*-Linked Parkinson’s Disease?

**DOI:** 10.3390/biom13010178

**Published:** 2023-01-15

**Authors:** Tatou Iseki, Yuzuru Imai, Nobutaka Hattori

**Affiliations:** 1Department of Neurology, School of Medicine, Juntendo University, 2-1-1 Hongo, Bunkyo-ku, Tokyo 113-8421, Japan; 2Department of Research for Parkinson’s Disease, Graduate School of Medicine, Juntendo University, 2-1-1 Hongo, Bunkyo-ku, Tokyo 113-8421, Japan; 3Research Institute for Diseases of Old Age, Graduate School of Medicine, Juntendo University, 2-1-1 Hongo, Bunkyo-ku, Tokyo 113-8421, Japan; 4Center for Genomic and Regenerative Medicine, Graduate School of Medicine, Juntendo University, 2-1-1 Hongo, Bunkyo-ku, Tokyo 113-8421, Japan; 5Neurodegenerative Disorders Collaborative Laboratory, RIKEN Center for Brain Science, 2-1-Hirosawa, Wako-shi, Saitama 351-0198, Japan

**Keywords:** leucine rich-repeat kinase 2, Parkinson’s disease, astrocytes, microglia, inflammation, glial cells, lysosomes

## Abstract

Leucine rich-repeat kinase 2 (*LRRK2*) is the most well-known etiologic gene for familial Parkinson’s disease (PD). Its gene product is a large kinase with multiple functional domains that phosphorylates a subset of Rab small GTPases. However, studies of autopsy cases with LRRK2 mutations indicate a varied pathology, and the molecular functions of LRRK2 and its relationship to PD pathogenesis are largely unknown. Recently, non-autonomous neurodegeneration associated with glial cell dysfunction has attracted attention as a possible mechanism of dopaminergic neurodegeneration. Molecular studies of LRRK2 in astrocytes and microglia have also suggested that LRRK2 is involved in the regulation of lysosomal and other organelle dynamics and inflammation. In this review, we describe the proposed functions of LRRK2 in glial cells and discuss its involvement in the pathomechanisms of PD.

## 1. Introduction

Leucine rich-repeat kinase 2 *(LRRK2*) is the causative gene for *PARK8*-linked Parkinson’s disease (PD) and the most common etiologic gene for familial PD [1,2,3]. Among the various mutations of LRRK2, G2019S mutations are found in 4% of familial and 1% of sporadic PD cases [4]. Other missense mutations linked to PD pathogenesis have also been identified throughout the LRRK2 protein (Figure 1). Although *LRRK2* is the most important causative gene of familial PD, its pathogenic mechanism remains unresolved. Autopsy brain studies of families with PD associated with LRRK2 mutations have shown the accumulation of TAR DNA binding protein of 43 kDa and tau inclusions with or without Lewy pathology [5], suggesting that the pathology may differ even within the same family [3]. Glial cell inclusions and progressive supranuclear palsy-like tau inclusions in the glia have also been reported [5,6]. Considering the multiple pathological findings including non-α-synuclein pathology and glial pathology in *LRRK2*-linked PD, a broad approach taking into account the whole brain environment may be required in addition to research focused on neurons and α-synuclein aggregation/propagation.

In the brain, LRRK2 is widely expressed in the neurons of the cortex, striatum, hippocampus, and substantia nigra [8,9], as well as in the microglia and astrocytes [10,11]. Moreover, LRRK2 shows a higher expression in inflammatory microglia than in resting-state microglia [10]. Several studies have proposed promising models of non-autonomous neuronal cell death caused by altered LRRK2 function in glial cells [12,13,14,15,16]. In this review, we summarize the glial phenotypes and altered organelle dynamics in glia caused by LRRK2 mutations and discuss the pathological mechanism of LRRK2-mediated non-autonomous neuronal cell death.

## 2. Structure and Function of LRRK2

LRRK2 is a large protein with a molecular weight of approximately 290 kDa (2527 amino acids) [3] and contains multiple domains, including an armadillo domain, an ankyrin domain, a Ras of complex protein (ROC) domain, a carboxyl-terminal of ROC (COR) domain, a kinase domain, and a WD40 domain (Figure 1). Due to the presence of both ROC and COR domains, LRRK2 and its paralog, LRRK1, belong to the ROCO family of proteins [17]. However, LRRK1 does not have an armadillo domain, which makes this domain unique to LRRK2. Rab29, a possible activator of LRRK2, and Rab10, a known LRRK2 substrate, are assumed to bind to the armadillo domain [18,19]. Depending on the cell context, LRRK2 exists as a monomer or as a dimer, wherein the dimer formation is mediated by binding via the ROC-COR and WD40 domains [20,21]. LRRK2 is mostly distributed in the cytoplasm as well as in vesicular membranes, such as endolysosomes [8,22,23] and synaptic vesicles [24,25,26], and regulates vesicular trafficking [27,28], mitochondrial homeostasis [29,30,31], and lysosomal function [11,22]. The loss of LRRK2 has been shown to alter the lysosomal structures in alveolar epithelial cells and urothelial cells in mice [32,33]. In primates, LRRK2 kinase inhibitors induced reversible histological changes, including vacuolation of the lung [34]. Additionally, the lungs and kidneys, where the obvious degeneration phenotypes are observed, show a strong LRRK2 expression [35]. Fibroblasts derived from patients with G2019S-associated PD also display lysosomal aggregation and enlargement [36]. In addition to the lysosomal function and trafficking, LRRK2 was revealed to be involved in autophagy. A study using cultured neurons demonstrated that LRRK2 G2019S or its upstream regulator Rab29 enhances JIP4 recruitment to autophagic vesicles via the phosphorylation of Rab10 and activates the kinesin motor, thereby disrupting the retrograde transport and maturation of autophagic vesicles [37]. LRRK2 is also reported to be a substrate for chaperone-mediated autophagy (CMA), which is disrupted by the I2020T and G2019S mutations [38].

LRRK2 was reported to show microtubule-associated localization, forming a skein-like structure [39,40]. Cryo-electron microscopic analysis further revealed that LRRK2 is polymerized on the surface of microtubules in a helical fashion [41,42]. Furthermore, LRRK2 was shown to be associated with non-acetylated microtubules [43]. In *Drosophila*, LRRK2 pathogenic mutants in the ROC-COR domain exhibited a strong interference in axonal transport, which was reversed by the promotion of microtubule acetylation [43]. In vitro studies showed that a truncated LRRK2 with the ROC-COR, kinase, and WD40 domains inhibited the motility of both kinesin and dynein motors on the microtubules in a concentration-dependent manner [41]. Moreover, LRRK2 interfered with primary cilia formation in a kinase-dependent manner by enhancing the binding of phospho-Rab10 with its effector, Rab-interacting lysosomal protein-like protein [44,45]. Additionally, Hedgehog signaling, which is sensed by the primary cilia, was altered in LRRK2 G2019 and R1441C knock-in mice, suggesting that the synaptic activity and survival of the striatal neurons and astrocytes are compromised by cilia loss [45,46].

The ROC domain has a Rab-like small GTPase structure and functions as a GTPase switch to regulate LRRK2 kinase activity [47,48]. Several phosphorylation targets of LRRK2 have been determined, which include Rab3A/B/C/D, Rab8A/B, Rab10, Rab12, Rab35, and Rab43 as endogenous substrates, Rab5B/C and Rab29 as potential substrates [19,44,49,50,51,52,53], along with nuclear translocation of nuclear factor of activated T-cells 1 (NFATc2/NFAT1), an inflammation-related transcription factor [14], and Wiskott–Aldrich syndrome protein family verprolin-homologous protein 2 (WAVE2), a regulator of actin dynamics [54]. Rab29, the gene of which is located in the *PARK16* locus [55], was characterized as an upstream molecule of LRRK2, recruiting LRRK2 to the vesicle membrane [56,57]. However, the involvement of Rab29 in LRRK2 signaling is under debate because LRRK2 was also activated in *Rab29* knock-out cells [58]. Many Rab proteins are involved in vesicular trafficking, such as synaptic dense core vesicles, endolysosomes, and trans-Golgi network. Thus, the alteration of vesicular trafficking may be a major pathogenic mechanism of LRRK2 mutation-associated PD [59]. 

## 3. Mammalian Model Studies of LRRK2

Studies of various transgenic LRRK2 mouse models revealed a possible role for LRRK2 in the dysfunction of striatal neurotransmission and dopaminergic homeostasis, yet the results are not consistent. For example, studies of mice overexpressing LRRK2 G2019S or R1441G in dopaminergic neurons showed reduced dopamine production or release and loss of the midbrain dopaminergic neurons [60,61]. Others reported an altered morphology of dopaminergic neurons without neuronal loss [62,63]. These differences could arise from the insertion loci of LRRK2 transgenes or the expression levels of the transgene itself, which may confound the LRRK2 phenotypes observed in different models and hinder our understanding of the roles of LRRK2 expressed at physiological levels. On the other hand, LRRK2 knock-in models, which are expected to have physiological levels of expression and distribution of gene expression, show only minor overall abnormalities in neurological function, limiting their value as disease models [64,65]. Surprisingly, LRRK2 knockout mice and rats show abnormalities in the lungs and kidneys, and the role of LRRK2 in non-neural tissues is currently being elucidated [64,65]. Importantly, in one study, microglial activation was suppressed in LRRK2 knockout rats, making them resistant to toxicity induced by α-synuclein and lipopolysaccharide (LPS) [66]. This observation strongly suggests a role for LRRK2 in glial cells’ physiology. Furthermore, LRRK2 is highly expressed in monocytes and macrophages, and the brain infiltration of macrophages differentiated from monocytes could contribute to pathology [10,67,68,69]. After birth, macrophages’ and microglia’s developmental trajectory diverges and the two populations may play different roles in the brain [70]. However, the contribution of monocytes and macrophages to LRRK2 pathology cannot be ignored. Thus, this review will also mention studies on these cells along with those on astrocytes and microglia (Figure 2).

## 4. Astrocytes

### 4.1. Astrocytes and PD

Astrocytes are the major glial cells in the central nervous system and are essential for providing the appropriate environment for neurons in the adult brain [73]. Astrocytes’ roles include the metabolic exchange with neurons, the regulation of neurotransmitter cycle in the synaptic cleft, and the maintenance of the blood–brain barrier [74,75]. Recently, changes in the astrocytic function have been suggested to lead to a pathological environment in several neurodegenerative diseases, such as Alzheimer’s and Huntington’s diseases [76,77]. Astrocytes are also of interest in PD because of the postulation of inflammatory astrocyte involvement in PD neurodegeneration [78,79,80]. The astrocytic expression of PD causative genes, including LRRK2, also strongly suggests the involvement of astrocytes in PD pathogenesis [81].

### 4.2. Role of LRRK2 in Astrocytes

As mentioned in Section 2, LRRK2 may regulate Rab-mediated vesicular trafficking. In mouse primary astrocytes, the lysosome number and size are dependent on LRRK2 kinase activity [11]. Lysosomal hypertrophy was observed in the presence of LRRK2 G2019S, R1441C, and Y1699C, which exhibit increased kinase activity [11]. Among the lysosome-associated molecules, mutations of Beta-glucocerebrosidase 1 (GBA1) have been identified as a risk factor for PD [82,83]. Lysosomal alkalinization and dysfunction were observed in primary astrocytes from GBA1 D409V knock-in mice, which were improved by the treatment with an LRRK2 inhibitor, suggesting that GBA1 is a lysosomal target of LRRK2 [84]. Lysosomal membrane permeabilization by l-leucyl-l-leucine methyl ester (LLOME), a lysosomotropic reagent, recruited LRRK2 to ruptured lysosomes in primary astrocytes, where activated LRRK2 phosphorylated Rab10 and recruited the motor adaptor protein JIP4 to the microtubules, leading to lysosomal tubulation and budding [23]. Although the physiological significance of this phenomenon is unclear, lysosomal tubulation and budding may act as a protective response by sorting undigested proteins to other active lysosomes for proper degradation [85]. Microtubule-dependent lysosomal positioning is determined by the kinesin motor-associated BLOC-one-related complex and Arf-like small GTPase (ARL8B) [86] and JIP4, the dynein motor-associated protein [87]. The movement of lysosomes to the perinuclear region is linked to autophagy activation [88] and also occurred in response to LLOME treatment [89]. The positioning of LRRK2-positive lysosomes determined the substrate specificity of Rab GTPase [89]. In astrocytes and HEK293FT cells, the recruitment and phosphorylation of Rab10 by LRRK2 only occurred in the perinuclear lysosomes. In contrast, Rab12 was recruited and phosphorylated by LRRK2 in both the perinuclear and the peripheral lysosomes [89]. Although PPM1H, the phosphatase of Rab GTPases, may be involved in this phenomenon, the detailed mechanism requires further analysis [90].

The role of LRRK2 in vesicular trafficking has been reported in a variety of experimental astrocyte systems, which suggest that both α-synuclein-associated and -non-associated mechanisms are involved in non-autonomous neuronal cell death. Endolysosomal dysfunction and impaired clearance of α-synuclein due to decreased phagocytosis-related annexin A2 have been demonstrated in primary astrocytes harboring LRRK2 G2019S [91]. In a study using induced pluripotent stem (iPS) cells derived from patients with LRRK2 G2019S-associated PD, the accumulation of α-synuclein in astrocytes due to impaired autophagy was also observed [12]. In a co-culture of LRRK2 G2019S astrocytes with dopaminergic neurons from healthy controls, astrocyte-derived pathogenic α-synuclein was transported into the dopaminergic neurons, resulting in neurodegeneration [12].

Changes in extracellular vesicles (EVs)/multivesicular bodies (MVBs) have also been observed in LRRK2 G2019S astrocytes derived from human iPS cells. Compared to normal astrocytes, LRRK2 G2019S astrocytes have smaller MVBs, accompanied with the accumulation of phospho-S129 α-synuclein and a lower EV release, while astrocyte–dopaminergic neuron co-culture systems have shown that the alteration of EV biogenesis in LRRK2 G2019S astrocytes induced neurite shortening and cell death in dopaminergic neurons [13]. Another human iPS cell study reported that LRRK2 G2019S astrocytes from patients with PD have reduced mitochondrial function and impaired neuronal interactions, mainly, synaptic interactions [92].

In summary, LRRK2 function in astrocytes may be closely related to organelle dynamics (Figure 3). Additionally, Rab proteins may very likely play an important role because of their downstream location. In LRRK2 pathological mutations, non-autonomous neuronal cell death may also be associated with altered organelle dynamics, suggesting that research on this mechanism may lead to a better understanding of PD pathogenesis. Recent single-cell analyses suggest that astrocytes may differ in function and properties in different brain regions [93,94]. Elucidating the possibility that LRRK2 may be of particular importance for local astrocytes in the substantia nigra and striatum will be an important future challenge.

## 5. Microglia

### 5.1. Microglial Cells and PD Etiology

Inflammation-responsive microglia with human leukocyte antigen-DR isotype have been observed in the brain pathology of PD; the extent to which reactive microglia contribute to PD pathogenesis remains undetermined [95]. Microglial innate immune signaling in mouse models of Alzheimer’s disease was found to cause synapse loss [96,97]. A pathway enrichment analysis from a meta-genome-wide association study (meta-GWAS) suggested at least some immune component involvement in PD development risk [98]. Considering these findings, the relation between PD etiology and immune responses is still a debated topic.

Cx3cr1-, CD11b-, Iba1-, and F4/80-positive microglial cells are a subset of glia that occupy approximately 10% of the brain. Microglia are resident intracranial immune cells with functions similar to those of macrophages [99]. They are mesenchymal and myeloid cells derived from the yolk sac and have self-renewal capability [100]. Physiologically, their main functions are sensing and housekeeping of the surrounding environment and protection against toxic substances [101]. Microglial uptake and degradation of extracellular α-synuclein aggregates is the most effective among all central nervous system cells in vitro, suggesting that microglia are major clearance cells for α-synuclein [4]. Extracellular α-synuclein released from neurons may be phagocytosed via toll-like receptor (TLR) 4, wherein TLR4 transcriptionally upregulated p62/SQSTM1, an autophagy receptor, via nuclear factor (NF)-κB signaling to lead to the degradation of α-synuclein by the autophagy–lysosomal pathway in microglia [102]. Although TLR4 exerted protective effects on dopaminergic neurons in terms of α-synuclein degradation, the TLR2–MyD88–NF-κB pathway was shown to promote α-synuclein propagation [103]. TLR2 recognized α-synuclein preformed fibrils because they could be a strong ligand of TLR2 [103]. α-synuclein preformed fibrils then activated the microglial TLR2–MyD88–NF-κB pathway and stimulated the secretion of proinflammatory cytokines, such as interleukin (IL)-1β and tumor necrosis factor-α (TNF-α) [103]. These microglia-derived proinflammatory molecules in turn transcriptionally upregulated α-synuclein levels in neurons, promoting α-synuclein propagation [103]. Conversely, the blocking of the TLR2 cascade was shown to ameliorate neuroinflammation and prevent α-synuclein fibril propagation in a mouse model of α-synuclein propagation [103]. This differential responsiveness of microglia to physiological and pathogenic α-synuclein via TLR2 and TLR4 suggests that microglia are involved in the pathological mechanisms of PD. Moreover, the activation of NF-κB signaling, which plays a central role in immune responses, was suggested to be protective in neurons and to contribute to neurotoxicity in glia [104]. These contradictory observations may be due to a focus on different effects of the NF-κB pathway.

### 5.2. Role of LRRK2 in Microglia

LRRK2 has been implicated in inflammation because *LRRK2* gene variants are linked to autoimmune diseases, such as Crohn’s disease and systemic lupus erythematosus [105,106]. LRRK2 is abundantly expressed in innate immune cells, including monocytes and neutrophils [107], where its expression is upregulated by interferon (IFN) γ [108,109]. At the cellular level, the amplification of inflammatory signals via LRRK2 mutations was observed in microglia as well as in monocytes and macrophages, with similar mechanisms [10,110]. Inflammatory cytokines are elevated in sporadic and *LRRK2*-linked PD [111]. Moreover, a correlation between disease severity and elevated levels of inflammatory cytokines was observed in the patients with *LRRK2*-linked PD [112]. 

Single-cell analysis also revealed a variant in the 5′ untranslated region of the *LRRK2* gene isolated in a GWAS of sporadic PD that was enhanced by microglia-specific LRRK2 expression [113]. Human iPS cell-derived monocytes and macrophages with LRRK2 G2019S were shown to potentiate the production of inflammatory cytokines, such as IL-1β, TNF-α, and IFNγ, following the activation of TLRs in a LRRK2 kinase activity-independent manner [110]. IFNγ also increased LRRK2 expression in neurons [15]. Moreover, conditioned medium from LPS-activated LRRK2 G2019S microglia, in which the nuclear translocation of NFATc2/NFAT1 was impaired, suppressed neurite outgrowth in both normal and G2019S neurons [15]. Protein kinase A-mediated phosphorylation of p50, the NF-κB inhibitory subunit, was negatively regulated by LRRK2, thereby contributing to the production of IL-1β in microglia [114].

Amoeboid-like immune cells, such as microglia and macrophages, are constantly changing their morphology in response to foreign entities. Considering that the lack of GbpC, a ROCO family protein, affects chemotaxis in *Dictyostelium,* an amoeboid organism, the ROCO-family proteins may be involved in the functions and motility of amoeboid-like cells [115]. In *Dictyostelium*, the cGMP-binding GbpC is involved in the chemotaxis mechanism via phosphorylation of myosin II [116]. 

LRRK2 phosphorylated and stabilized WAVE2, a regulator of the actin cytoskeleton, leading to a positive regulation of phagocytosis in microglia [54]. LPS-mediated activation of microglia has been demonstrated to promote dopaminergic neuron death via the LRRK2–WAVE2 pathway [54]. Apart from the positive regulation of phagocytosis via WAVE2 phosphorylation, LRRK2 was also shown to negatively regulate the migration of microglia through the suppression of focal adhesion kinase via phosphorylation [117]. These results are indirectly supported by the observation that CX3CL1, which activates microglial motility, showed increased levels in LRRK2-null microglia [118]. In contrast to that observed in microglia, the inhibition of LRRK2 kinase attenuated chemotaxis to α-synuclein fibrils in primary cultured macrophages and the infiltration of peripheral monocytes into the central nervous system [69]. Moreover, the PD-linked LRRK2 G2294R mutation in the WD40 domain, which destabilized the LRRK2 protein, resulted in an attenuated macrophage uptake of α-synuclein fibrils [119]. 

In the study analyzing the inflammatory cascade, NFATc2 was reported to be phosphorylated by LRRK2 in microglia exposed to α-synuclein and translocated into the nucleus for transcriptional regulation, and this mechanism was inhibited in the primary microglia of *LRRK2* knock-out mice [14]. However, another study of iPS microglia derived from patients with the G2019S mutation showed inhibition of the nuclear migration of NFATc2 [15]. Additionally, the study also reported that the microtubule-dependent nuclear translocation of NFATc4 was impaired in neurons, leading to defects in neurite outgrowth [15].

LRRK2 pathogenic mutants were demonstrated to sequester Rab8 to the lysosomes via phosphorylation, directing the transferrin-mediated iron endocytosis pathway in microglia from recycling to degradation [16]. Mislocalization of transferrin at the lysosomes was also observed in LPS-induced inflammatory microglia [16]. Moreover, iron and ferritin deposition in inflammatory microglia was observed in the striatum of G2019S knock-in mice [16]. 

In the macrophage cell line RAW264.7, LRRK2 accumulated and recruited Rab8 and Rab10 under lysosomal stress in a Rab29-dependent manner, which led to the extracellular release of the lysosomal contents [22,23]. LRRK2 was also mobilized to the lysosomes damaged by LLOME. Moreover, Rab10-mediated lysosomal tubulation and budding to the plasma membrane were detected in astrocytes [23]. Furthermore, lysosome enlargement due to LRRK2 inhibition in RAW264.7 cells suggested that LRRK2 is involved in lysosome quality control, as proposed in the previously mentioned astrocyte study that revealed LRRK2-dependent lysosomal budding [22].

LRRK2 was also translocated to maturing phagosomes in human iPS cell-derived macrophages in a kinase activity-independent manner [109]. Although LRRK2 was not involved in the initial phagocytosis, LRRK2 kinase activity was required for the mobilization of Rab8a and Rab10 to the phagosomes, suggesting that LRRK2 is required in the phagosome maturation pathway [109].

Microglia exchange mitochondria and materials through F-actin-associated nanotubes [31]. Microglia that had taken up excess amounts of α-synuclein fibrils transported these fibrils to other microglia through F-actin-associated nanotubes, while inflammatory microglia accepted healthy mitochondria from naive microglia to improve survival and diminish the inflammatory profile [31]. In this context, LRRK2 G2019S disturbed this transport in a kinase-activity-independent manner, which could contribute to the exacerbation of inflammation by α-synuclein fibrils [31].

In addition to the abnormalities in vesicle trafficking and lysosomal function, microglial studies indicated a link between LRRK2 and both inflammation and phagocytosis (Figure 4). As noted in the section on astrocytes, microglia may also be involved in non-autonomous neuronal cell death.

## 6. Conclusions and Perspective

We reviewed LRRK2 studies in glial cells, wherein LRRK2 is suggested to have diverse roles in non-autonomous neuronal cell death/neuroinflammation. The dysregulation of organelle dynamics, including endolysosomes, was demonstrated in LRRK2-mutated astrocytes, while the disturbance of the inflammatory cascade and phagocytosis/migration mechanism has been proposed in LRRK2-mutated microglia in addition to the dysfunction of organelle dynamics and mitochondria. In contrast, the pathological analyses of PD brain with LRRK2 R1441H [120] and G2385R [121] indicated that the inflammatory profile is not as prominent as observed in the basic experiments. Although these results can be attributed to the fact that the pathological analysis studies may have detected results after the inflammatory response had terminated (at the time when neuronal death was already complete), this issue is still unresolved.

Recently, parenchymal border macrophages have been reported to regulate the flow dynamics of the cerebrospinal fluid [122]. Given the diversity of the LRRK2 pathology, the disruption of the glymphatic system by macrophages with LRRK2 mutations may result in an impaired efflux of aberrant proteins [123].

Phenotypic analyses of *LRRK2* knock-out animals showed degeneration of the renal proximal tubules and pulmonary epithelial cells [124], enlargement of lysosomes [125,126], altered axonal transport [127], and synaptic dysregulation [24,26]. In contrast, the nigrostriatal dopaminergic system was not functionally compromised in *LRRK2* knock-out mice [32] and human cases with *LRRK2* loss-of-function variants [128]. LRRK2 kinase inhibitors [129] are currently being investigated based on the results that the causes of human *LRRK2* loss of function are not strongly associated with any specific phenotype or disease state [128]. Especially, a phase III trial has begun for an LRRK2 inhibitor, BIIB122, that is being developed by Denali and Biogen [130]. This drug is the most advanced Parkinson’s disease modifying drug in development. However, considering that the broad physiological role of LRRK2 is potentially not limited to kinase activity, further analysis may be needed for a more optimal drug development.

## Figures and Tables

**Figure 1 biomolecules-13-00178-f001:**
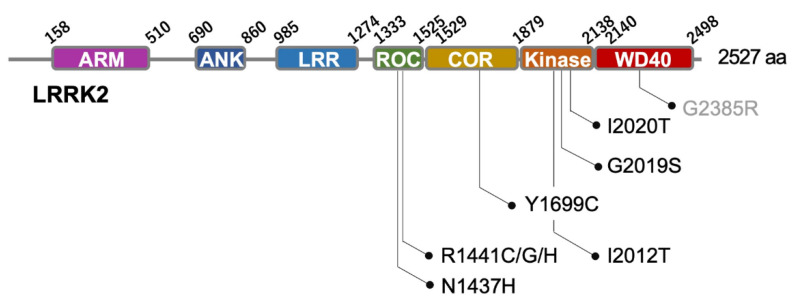
Domain structure of LRRK2. Sites of representative pathological mutations are indicated [7]. G2385R is considered a risk variant in Asian races [7]. LRRK2, leucine rich-repeat kinase 2; ARM, armadillo domain; ANK, ankyrin repeat domain; LRR, leucine-rich repeat domain; ROC, Ras of complex protein domain; COR, carboxyl-terminal of ROC domain; WD40, WD 40 domain; aa, amino acids.

**Figure 2 biomolecules-13-00178-f002:**
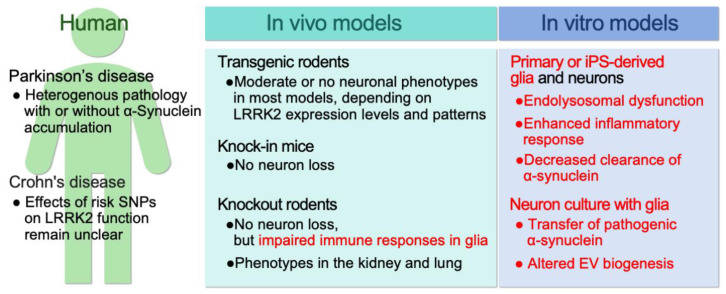
Overview of LRRK2 research, research problems, and issues to be resolved. In humans, risk SNPs have been found in Crohn’s disease as well as Parkinson’s disease. Currently, the impact of Crohn’s disease-associated SNPs on LRRK2 function is not clear [71,72]. Many in vivo models do not exhibit the pathological changes observed in Parkinson’s disease. However, LRRK2 knockout rodent models are an excellent tool to study the physiological function of LRRK2. In vitro models are superior for analysis at the organelle and molecular levels but have the disadvantage of not being able to reproduce the normal aging process and the brain environment. The results obtained from each of these models should be assessed in light of these considerations. The studies to which this review refers are indicated in red [11,12,13,14,73]. *C. elegans* and *Drosophila* models are excellent for molecular genetic analysis but will not be mentioned here. EV, extracellular vesicle.

**Figure 3 biomolecules-13-00178-f003:**
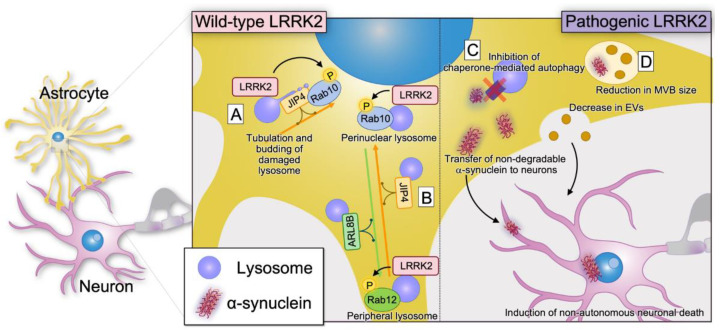
Reported roles of LRRK2 in astrocytes. (**A**) LRRK2 accumulates in membrane-damaged lysosomes and phosphorylates downstream Rab10, causing lysosomal tubulation and budding [23]. (**B**) Microtubule-dependent kinesin–ARL8B and dynein–JIP4 motors move the lysosomes to the peripheral and perinuclear regions, respectively [89]. Rab10 is preferentially phosphorylated in the perinuclear lysosomes by LRRK2 (**A**), whereas Rab12 is phosphorylated in both the perinuclear and the peripheral lysosomes. (**C**) Impairment of chaperone-mediated autophagy by LRRK2 G2019S leads to α-synuclein accumulation in astrocytes [12]. Astrocyte-derived pathogenic α-synuclein is transported into dopaminergic neurons [12]. On the other hand, phagocytic defects of α-synuclein fibrils due to decreased annexin A2 by LRRK2 G2019S are also suggested [91]. (**D**) Compared to normal astrocytes, LRRK2 G2019S astrocytes have smaller MVBs, along with the accumulation of phospho-S129 α-synuclein and lower EV release. The alteration of EV biogenesis in LRRK2 G2019S astrocytes induces dendritic shortening and cell death in dopaminergic neurons [13]. MVB, multivesicular body; EV, extracellular vesicle; P, phosphorylation.

**Figure 4 biomolecules-13-00178-f004:**
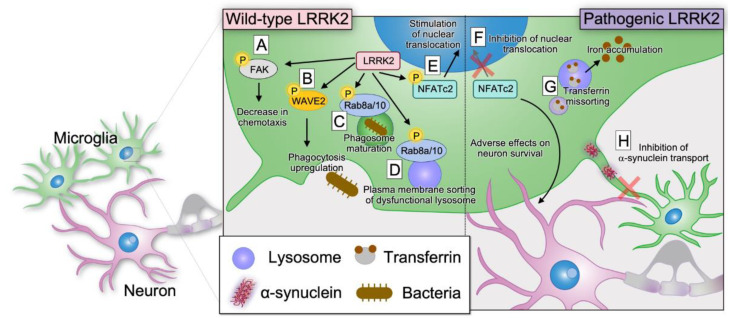
Reported roles of LRRK2 in microglia. (**A**) LRRK2 negatively regulates the migration of microglia through the suppression of focal adhesion kinase (FAK) via phosphorylation [117]. (**B**) LRRK2 phosphorylates and stabilizes WAVE2, inducing the positive regulation of microglial phagocytosis [54]. (**C**) LRRK2 kinase activity is required for the mobilization of Rab8a and Rab10 to phagosomes, which leads to phagosome maturation [109]. (**D**) LRRK2 accumulates and recruits Rab8 and Rab10 under lysosomal stress in a Rab29-dependent manner, which causes the extracellular release of the lysosomal contents [22]. (**E**) NFATc2 is phosphorylated by LRRK2 in microglia exposed to α-synuclein and then translocated to the nucleus for transcriptional regulation [14]. (**F**) Nuclear migration of NFATc2 is inhibited in LRRK2 G2019S iPS cell-derived microglia [15]. (**G**) LRRK2 pathogenic mutants sequester Rab8 to the lysosomes via phosphorylation, directing the transferrin-mediated iron endocytosis pathway from lysosomal recycling to degradation [16]. As a result, iron deposition is accelerated [16]. (**H**) Microglia that have taken up excess amounts of α-synuclein fibrils transport these fibrils to other microglia through F-actin-associated nanotubes [31]. P, phosphorylation.

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
