# Peer review of "Is Glial Dysfunction the Key Pathogenesis of LRRK2-Linked Parkinson’s Disease?"

_biomolecules, 2023, doi:10.3390/biom13010178_

Round 1

Reviewer 1 Report

This is an informative review of the role of glial cells in LRRK2 -linked PA. The topic is interesting and the information provided is rich.

However, it remains unclear whether glial dysfunction is actually playing the key role in the pathology. This is mainly due to the fact that the authors did not juxtapose the role of LRRK2 in neurons to that in glial cells. In fact, the review highlights mechanistic aspects of LRRK2 that might be shared between glial cells and neurons.

For example, it is not clearly discussed of whether astrocytes dysfunctions are solely driven by mutations in LRRK2 or are also a consequence of neuronal dysfunction or both. It remains also unclear of whether astrocytes, microglia or non-glial immune cells are involved in actively inducing neuronal dysfunctions. The authors eluded to this topic in line 154-166 but do not clearly discuss the important implications of what cell population might represent the primary target cells in PA. In addition, the authors could have included a discussion of the timeline of pathology and the role of astrocytes and microglia in disease progression. Any discussion along these lines is missing and makes the review less informative and novel.

In addition, it would have been interesting to add a discussion of the specificity of astrocytic dysfunction to PA in general. For example, astrocyte-specific deletion of other genes i.e Sumf1 has been shown to also lead to lysosomal storage and autophagy dysfunction with consequential cytoplasmic accumulation of autophagic substrates. However, these Sumf1 associated lysosomal dysfunctions in astrocytes are not associated with development of PA but lead to very different neurodegenerative disorders.  It would have been interesting to discuss how different lysosomal functions in astrocytes are associated with entirely different neurodegenerative disorders and why mutation in LRRK2 lead to the very specific PA disease. In addition, a discussion of the regional astrocyte dysfunction that might explain the vulnerability of dopamine neurons is not well developed.

Apart from the missed opportunity to add more interesting aspects to astrocyte d=dysfunctions, the section of microglia (4.2) is quite confusing as the authors intermix microglia related observation with macrophages and monocytes. As macrophages are often recruited from the periphery, (and are not glial cells) observation on microglia should be separated from observation of other immune cells.

In addition, the inclusion of other immune cells in the review is not captured by the title and distracts from the role of LRRK2 in glial cells. The reader is left with wondering whether glial defects are a secondary consequence of immune activation driven by invading macrophages or monocytes.

A balance review would also provide some data that might NOT support the overall stated question. Such data should be included.

Finally, the overall review is dense and difficult to follow. Additional graphic schematics would help and would make the review more approachable.

Minor points:

Citation 10 does not describe expression of LRRK2 in microglia as suggested by the authors. Please correct this citation.

Line 155 should read “dopaminergic neurons FROM healthy…”

Add and refer to Zhang et al (Front. Cell. Neurosci., 26 May 2022 Sec. Cellular Neuropathology
https://doi.org/10.3389/fncel.2022.903469) which is a review on the same topic.

Reviewer 2 Report

Iseki and colleagues report in a compact and comprehensive review the recent updates about the role of LRRK2 in glial cells and, in turn, its possible involvement in pathomechanisms of PD, when mutated.

This review is well-written, easy to follow and clarifies efficiently the state of the art in this field.

I hereafter report some comments/suggestions for the authors to improve the manuscript:

1)     I would report (in Section 2, last paragraph) the role of Rab proteins in the dismissal of altered mitochondria. In general, mitochondria are neglected in the present review. Since they are central in the pathogenetic process, I would add some references about the LRRK2-mitochondria axis in general and LRRK2/mitochondria/glia in particular (for example in Scheiblich H, Dansokho C, Mercan D, Schmidt SV, Bousset L, Wischhof L, Eikens F, Odainic A, Spitzer J, Griep A, Schwartz S, Bano D, Latz E, Melki R, Heneka MT. Microglia jointly degrade fibrillar alpha-synuclein cargo by distribution through tunneling nanotubes. Cell. 2021 Sep 30;184(20):5089-5106.e21. doi:10.1016/j.cell.2021.09.007. Epub 2021 Sep 22. PMID: 34555357; PMCID: PMC8527836.)

2)     I suggest the authors to dedicate some comments (chapter 4.1) to the role of NF-kB in PD pathogenesis. It seems (but this is my personal opinion based on literature) that this pathway is neuroprotective, when activated in neurons, and neurotoxic, when activated in glial cells. This discrepancy may give rise to non-conclusive (or opposite) results in different papers.

3)     In “conclusions and perspective”, authors just cite studies on LRRK2 inhibitors. To my knowledge, they are the only (together with GCase activators) drugs currently being investigated in clinical trials (e.g., DENALI, DENALI+Biogen) as disease-halting therapies. In my opinion, it is worth stressing this point.

Minor point:

Line 168 Rab should be Rab proteins or Rabs, since authors are referring to several Rabs.

Round 2

Reviewer 1 Report

The revised version is highly improved  and the newly added Figure is very helpful. The authors were overall highly responsive and addressed all concerns and questions in an effective manner . Overall, this is an informative and balanced review of the stated topic and will be of interest to a wide audience.